# SAMPLE-EFFICIENT IMITATIVE MULTI-TOKEN DECISION TRANSFORMER FOR REAL-WORLD DRIVING

## ABSTRACT

Recent advancements in autonomous driving technologies involve the capability to effectively process and learn from extensive real-world driving data. Current imitation learning and offline reinforcement learning methods have shown remarkable promise in autonomous systems, harnessing the power of offline datasets to make informed decisions in open-loop (non-reactive agents) settings. However, learning-based agents face significant challenges when transferring knowledge from open-loop to closed-loop (reactive agents) environment. The performance is significantly impacted by data distribution shift, sample efficiency, the complexity of uncovering hidden world models and physics. To address these issues, we propose Sample-efficient Imitative Multi-token Decision Transformer (SimDT). SimDT introduces multi-token prediction, online imitative learning pipeline and prioritized experience replay to sequence-modelling reinforcement learning. The performance is evaluated through empirical experiments and results exceed popular imitation and reinforcement learning algorithms both in open-loop and closed-loop settings on Waymax benchmark. SimDT exhibits 41% reduction in collision rate and 18% improvement in reaching the destination compared with the baseline method.

## 1 INTRODUCTION

The rapid advancement in machine learning has led to unprecedented achievements in the autonomous systems domain. One of the critical methodologies in training these neural networks involves leveraging large offline datasets. These datasets provide the foundational knowledge that neural networks require to learn and make predictions or decisions. However, deploying these offline-trained neural networks in real-world scenarios or simulators with reactive agents presents challenges stemming from the distribution shift between the dataset and the application environment.

Learning from Demonstration (Dauner et al., 2023) (Cheng et al., 2023) (Huang et al., 2023) (Levine et al., 2020) encounter hurdles when presented scenarios deviate from the training distribution, exemplified by rare events like emergency braking for unforeseen obstacles. Similarly, these methods grapple with long-tail distribution phenomena during closed-loop tests, such as navigating through unexpected weather conditions or handling the erratic movements of jaywalking pedestrians.

Interacting with environments allows for improved management of out-of-distribution issues in closed-loop tests by learning policies focused on reward maximization, thus facilitating adaptation. However, reinforcement learning (RL) faces challenges in overcoming the simulation-reality discrepancy and suffers from low sampling efficiency (Kiran et al., 2022) (Liu et al., 2022). Traditional reinforcement learning methods also grapple with the complexities of large state spaces, long-term planning, and sparse rewards, mirroring the challenges of real-world driving scenarios. Decision Transformer (Chen et al., 2021b) leverages a transformer-based architecture to learn policies for decision-making in reinforcement learning tasks via sequence modeling. Despite its potential to scale with large state space (Zhou et al., 2024), the original architecture and pipeline are designed for offline learning and are not enough for complex and dynamic autonomous driving tasks. Classic RL techniques like prioritized experience replay (Schaul et al., 2016) are designed for large-scale datasets but cannot be applied as the Decision Transformer does not compute temporal differences.

On the other hand, single-token prediction does not fully fit the nature of autonomous driving and has difficulty uncovering the hidden physical world model. Concentrating solely on single-token

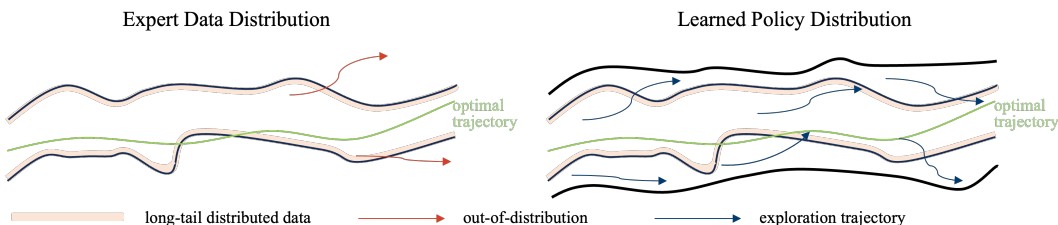

Figure 1: Comparative Illustration of Learning Approaches. The left figure depicts a data distribution of expert data, highlighting its limitations in managing distributional shifts and challenges arising from suboptimal training data. In contrast, the right figure presents our imitative reinforcement learning pipeline that demonstrates enhanced robustness by adapting policies online, thereby achieving superior performance under variable conditions.

prediction renders the model excessively susceptible to immediate contextual patterns, thereby neglecting the necessity for more extensive deliberation over protracted sequences. Models developed through next-token prediction techniques necessitate extensive datasets to attain a level of cognitive proficiency that humans achieve with significantly less exposure to tokens (Gloeckle et al., 2024).

This paper seeks to address the challenges associated with closed-loop autonomous driving by introducing Sample-efficient Imitative Multi-token Decision Transformer. The proposed approach leverages real-world driving data and realistic simulators, providing scenarios that are both realistic and conducive to online adaptation to distribution shifts. By employing multi-token prediction via sequence modeling of (state, action, return_to_go) pairs, the network is enabled to discern the quality of various action sequences, thereby gaining a deeper insight into the underlying world model. Experiment results indicate that our approach yields a substantial performance enhancement in terms of policy robustness and sample efficiency both in open-loop and closed-loop settings. SimDT exhibits 41% reduction in collision rate and 18% improvement in reaching the destination compared with the baseline method. The work is conducted entirely with Jax, which facilitates highly efficient training on large-scale data and real-time inference. The inference time is 1.63 milliseconds for SimDT(median) on RTX 3090. The main contributions are as follows:

- We present an online imitative reinforcement learning pipeline designed for wide data distribution across a collaboration of real-world driving datasets and simulators.
- We propose multi-token Decision Transformer architecture for receding horizon control to enhance long-horizon prediction and broaden the attention field.
- We introduce prioritized experience replay to sequence modeling-based reinforcement learning and enable sample-efficient training for large-scale data.

## 2 RELATED WORK

**Reinforcement learning via sequence modeling**. Trajectory Transformer (Janner et al., 2021) and Decision Transformer (DT) (Chen et al., 2021b) are pioneers in this area, leveraging transformer architectures to model sequences of state-action-reward trajectories and predicting future actions in an offline manner. Following work (Lee et al., 2022) Meng et al. (2022) (Wu et al., 2023) (Badrinath et al., 2023) extends leverage the power of transformers for efficient and generalized decision-making in RL. Online DT (Zheng et al., 2022) and Hyper DT (Xu et al., 2023) adapt the original concept for online settings and interact with environments. However, previous work are done on relatively simple environments compared to autonomous driving environments.

**Multi-token prediction**. Transformers have significantly impacted NLP since their inception (Vaswani et al., 2017), outperforming RNNs and LSTMs by processing sequences in parallel and efficiently handling long-range dependencies. Subsequent models like GPT (Radford et al., 2018) and BERT (Devlin et al., 2019) have refined the architecture, enhancing pre-training, fine-tuning, and scalability. Recent studies explore multi-token prediction on semantic representation (Kitaev et al., 2020), streamline computation (Wang & Cho, 2019), prediction technique (Qi et al., 2020) and multilingual (Jiang et al., 2020). However, focusing only on single-token prediction makes the model too sensitive to the immediate context and overlooks the need for deeper analysis of longer

sequences(Gloeckle et al., 2024). This paper extends the concept to Decision Transformer and explores the potential benefits of multi-token prediction for motion planning.

**Learning with real-world driving data.** Much work has been done to accommodate with real-world driving data (Sun et al., 2020)(H. Caesar, 2021)(Houston et al., 2020)(Phan-Minh et al., 2023) for generalizable driving policy. Lu et al. (2023) explores the cooperation between reinforcement learning and imitation learning in terms of loss design for real-world driving data. TuPlan (Dauner et al., 2023) combines both learning methods with rule-based methods for real-world planning. Guided Online Distillation (Li et al., 2023) adapts real-world data for reinforcement learning online distillation in MetaDrive Simulator (Li et al., 2022). Trajeglish (Philion et al., 2024) models real-world traffic autoregressively as a language processing problem with a causal transformer. Our approach is most similar to Trajeglish as both works tokenize expert driving logs to state-action sequences, encode agent and map information for better scene understanding and finally output actions for agent control. However, Trajeglish is fundamentally a supervised learning method and it inevitably has training and test distribution mismatch. Our approach differs as SimDT additionally models return to state-action sequences and convert supervised learning to reinforcement learning sequence modeling (state-action-return) problem to further solve the distribution shift issue. What's more, SimDT further develops online adaptation and multi-token prediction in contrast to Trajeglish's next-token prediction to further enhance its modeling of the driving data.

## 3 METHODS

In this section, we introduce SimDT, a novel sample-efficient reinforcement learning framework designed for dynamic driving scenarios through sequence modeling. SimDT integrates a multi-token decision transformer for nuanced decision-making, an online imitative reinforcement learning pipeline for continuous adaptation and improvement, and a prioritized experience replay mechanism to enhance learning efficiency by focusing on more informative experiences.

### 3.1 NETWORK STRUCTURE

Since the real-world driving environment is complex and dynamic, a specific feature encoding network is designed for the state representation. The real-world driving state contains perceptual information such as obstacles, road map, traffic and so on. We follow the vectorized representation to organize the road map as polylines and then extract with Polyline Encoder (Shi et al., 2022). Obstacles with past 10 historical information are recorded in terms of [position_x, position_y, yaw, speed, length, width]. Obstacle and traffic embedding are extracted with multi-layer perception network.

The work further extends the method to goal-conditioned reinforcement learning by adding the relative vector goal between the ego vehicle and the destination in terms of $(x, y)$. The importance of goal condition lies in its influence on the decision-making process of the autonomous agent. Even in an identical environment, the actions taken by the vehicle can vary significantly depending on the specified goal position.

Real-world data are recorded in terms of trajectory and it is converted to actions via inverse kinematics. Multi-token prediction is designed to enhance the network's understanding of reference trajectories and short-term momentum impact by learning a fixed horizon of reference actions (Figure 2,7). Multi-token prediction in a causal transformer simultaneously generates multiple actions in a single forward pass, while still respecting the autoregressive property that ensures each prediction only depends on previously generated tokens. This is typically achieved by using a masked self-attention mechanism that allows the model to consider multiple future positions without violating causal dependencies. As shown in equation 1, the next-token prediction has loss function $L_a$ defined as the negative log-likelihood of the policy.

$$L_a = -\log \pi_\theta(a_t \mid s_{t:t-c}, a_{t-1:t-c}, g_{t:t-c}) \tag{1}$$

where $\pi_\theta$ is the training driving policy. maximize the probability of $a_t$ as the next prediction action, given the history of past tokens with context length $c$ of $s_{t:t-c} = s_t, ..., s_{t-c}$. $a_{t-1:t-c} = a_{t-1}, ..., a_{t-c}$. $g_{t:t-c} = g_t, ..., g_{t-c}$.

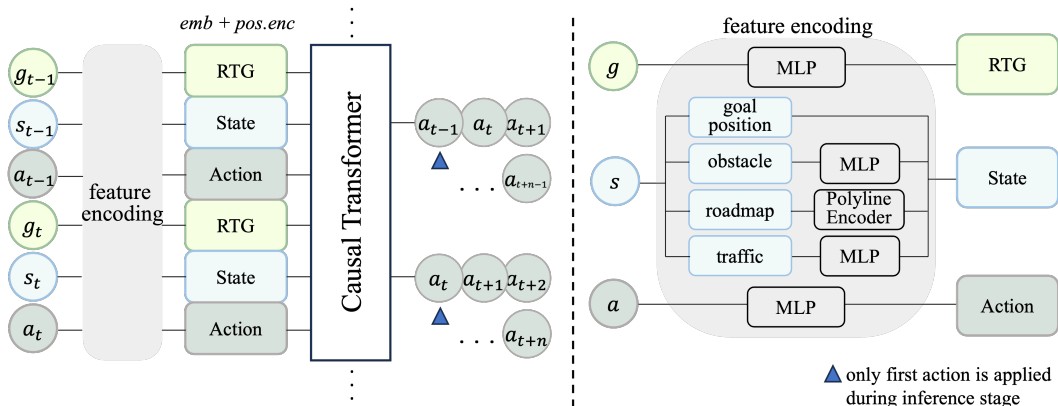

Figure 2: The general network architecture of SimDT. Feature encoding is applied to extract complex real-world driving perceptual data to small but meaningful embeddings. Inside the causal transformer, attention relationships are calculated for past context length of $[(s_t, a_t, g_t)]_{T-c}^{T}$. The decoder now predicts multi-tokens for actions and only the first action is applied during the inference stage.

The loss function is modified for the next n token prediction. Where $\alpha, \beta$ ... and $\gamma$ are the coefficients designed for the network to learn more about current step action predictions.

$$
\begin{aligned}
L_{ma} = &- \alpha * \log \pi_\theta(a_t \mid s_{t:t-c}, a_{t-1:t-c}, g_{t:t-c}) \\
&- \beta * \log \pi_\theta(a_{t+1} \mid s_{t:t-c}, a_{t-1:t-c}, g_{t:t-c}) - \gamma * \log \pi_\theta(a_{t+2} \mid s_{t:t-c}, a_{t-1:t-c}, g_{t:t-c}) \\
&- \quad \cdots\cdots \quad - \omega * \log \pi_\theta(a_{t+n} \mid s_{t:t-c}, a_{t-1:t-c}, g_{t:t-c})
\end{aligned}
\tag{2}
$$

### 3.2 ONLINE IMITATIVE REINFORCEMENT LEARNING PIPELINE

The general idea of the proposed Algorithm 1 is to perform online imitative reinforcement learning with off-policy expert data for pre-training at the beginning. Subsequently, the model undergoes a mixed on-policy adaptation phase which is introduced at the mid-point of the training process. The core concept behind this is to quickly shift the distribution towards the expert behavior at the beginning and reduce environmental distribution shift with on-policy adaptation (OPA). Note online policy adaptation and imitative reinforcement learning are performed concurrently after mid of training, this helps the network not to fall into online local minimal.

Imitative reinforcement learning is done by applying a similar concept as Shaped IL (Judah et al., 2014) and GRI (Chen et al., 2021a) where the reward is shaped for expert demonstration data. Following the same implementation in (Lu et al., 2023), expert data from real-world driving trajectory was converted to expert agent actions with inverse kinematics. We also design negative rewards for off-road and overlap (collision) behavior. The network learns good behavior through imitation rewards and bad actions through online interaction with off-road and overlap rewards. The overall online imitative reinforcement pipeline is essential to achieve the greater data-distributed policy described in Figure 1.

reward function:

$$
R_{imitaiton} = \begin{cases} 1.0 & \text{if log\_divergence} < 0.2, \\ 0.0 & \text{if log\_divergence} > 0.2, \end{cases}
\tag{3}
$$

$$
R_{off-road} = -2
\tag{4}
$$

$$
R_{overlap} = -10
\tag{5}
$$

where log_divergence is the euclidean distance between the log history of the ego vehicle and that of the controlled agent.

The real-time collected transition level replay buffer does not contain return-to-go as it can only be calculated after the episode is finished and all rewards are collected. Similar to Online DT(Zheng et al., 2022), the transition level replay buffer is converted to hindsight trajectory replay buffer when

---

**Algorithm 1** Online Imitative Reinforcement Learning Pipeline

---

Initialize Transition Replay Buffer $D_{trans}$ for capacity A, Trajectory Replay Buffer $D_{traj}$

**while** $n \leq num\_scenarios$ **do**

    **while** $D_{trans}$ is not full **do**                                    ▷ Online Data Collection

        **if** $n \leq 0.5 * num\_scenarios$ **then**

            reproduce scenarios with Human Expert Driving Data, $D_{trans} \leftarrow (s, a, r)$

        **else**

            reproduce scenarios with Human Expert Driving Data, $D_{trans} \leftarrow (s, a, r)$

            explore scenarios with Policy agent $\pi_\theta$ , $D_{trans} \leftarrow (s, a, r)$

        **end if**

    **end while**

    HindsightReturnRelabeling: $D_{traj} \leftarrow D_{trans} \equiv [[(s_{i,j}, a_{i,j}, g_{i,j})]_{i=0}^T]_{j=0}^{A/T} \leftarrow [(s_i, a_i, r_i)]_{i=0}^A$

    $D_{trans} \leftarrow \emptyset$

    **for** k in range(1000) **do**:

        sample and ShuffleObstacleOrder: $[[(s_{i,j}, a_{i,j}, g_{i,j})]_{i=t-c}^t]_{j=0}^B \leftarrow D_{traj}$

        train on sampled data

    **end for**

**end while**

---

a fixed amount of trajectories are collected (HindsightReturnRelabeling). Moreover, the dataloader randomly shuffles obstacles in order within the state (ShuffleObstacleOrder) to enhance data augmentation.

### 3.3 Prioritized Experience Replay for Decision Transformer

---

**Algorithm 2** Prioritized Experience Replay for Decision Transformer

---

Initialize Prioritized Trajectory Replay Buffers $D_{single}^{per}$, $D_{overall}^{per}$ with capacity $B$

**while** $n \leq$ num_scenarios **do**

    Execute online data collection from Algorithm 1                     ▷ Online Data Collection

    HindsightReturnRelabeling: $D_{traj} \leftarrow D_{trans} \equiv [[(s_{i,j}, a_{i,j}, g_{i,j})]_{i=0}^T]_{j=0}^{A/T} \leftarrow [(s_i, a_i, r_i)]_{i=0}^A$

    $D_{trans} \leftarrow \emptyset$

    **for** k in range(1000) **do**:

        Sample and ShuffleObstacleOrder: $[[(s_{i,j}, a_{i,j}, g_{i,j})]_{t-c}^t]_{j=0}^B \leftarrow D_{traj}$

        train on sampled data and obtain $L_{single}$ and $L_{overall}$

        $D_{single}^{per} \leftarrow \{[[(s_{i,j}, a_{i,j}, g_{i,j})]_{i=t-c}^t]_j, L_{single}\}$

        $D_{overall}^{per} \leftarrow \{[[(s_{i,j}, a_{i,j}, g_{i,j})]_{i=t-c}^t]_j, L_{overall}\}$

    **end for**

    Train on $D_{single}^{per}$ and $D_{overall}^{per}$

**end while**

---

The proposed imitative reinforcement learning method not only handles large volumes of offline demonstration data but also obtains an infinite amount of data through online interactions. The ability to utilize prioritized experience replay (PER) becomes critical for enhancing sample efficiency. The original proposed PER(Schaul et al., 2016) selectively samples experiences with high temporal-difference errors from the replay buffer for focusing on more informative experiences. However, DT does not use temporal-difference errors and therefore precludes direct application of PER. Instead, we adapt by using action loss to gauge transition importance within the Decision Transformer, which assesses state-action-return relationships. The design concept is that if the model's predicted actions diverge from actual ones, it indicates a misinterpretation of the environment.

We develop prioritized experience replay for Decision Transformer on top of the previous online imitative training pipeline(Algorithm 2). Extra replay buffers are designed to store prioritized sampled trajectories based on action loss. The action loss represents the difference between the actions predicted by the policy network and the actual actions taken. A low actor loss means that the policy network's predictions are close to the actual actions, while a high actor loss means that the predictions are far from the actual actions. Different from bootstrapping RL algorithms which sample

one pair of state, action and reward $(s_i, a_i, r_i)$, Decision Transformer samples a context length($c$) of state-action-return pairs $[[(s_{i,j}, a_{i,j}, g_{i,j})]_{i=t-c}^{t}]_j$ for sequence modeling. Therefore, the replay buffers designed for the decision transformer store a set of state-action-return pairs with a specified context length. Our proposed prioritization rule takes advantage of these state-action-return pairs, which try to analyze and understand either single scene or cumulative scenes. The proposed method prioritizes trajectories based on the following criteria:

**Criterion 1:** Preservation of transitions which contains single-step action discrepancy($L_{single}$): This methodology concentrates on isolating the instances where the model's prognostications manifest the greatest deviation from expected accuracy. Such a strategy is instrumental in directing the model's learning efforts toward ameliorating its most significant errors. The replay buffer for single-step action discrepancy $D_{single}^{per}$ can generally hold unexpected collisions and out-of-line actions.

**Criterion 2:** Preservation of transitions with maximal cumulative action discrepancy($L_{overall}$): This methodology is characterized by its emphasis on identifying and retaining sequences wherein the aggregate error of the model's predictions reaches its apex. It holds particular utility for endeavors aimed at refining the model's performance across a continuum of actions for context length of state-action-return pairs. The replay buffer for cumulative action discrepancy $D_{overall}^{per}$ is useful for preserving rare or long-tail scenarios.

The two replay buffers store data based on high value in low value out. The prioritized experience replay buffer is sampled for training every fixed amount of episode and its priorities are updated in the meantime. The goal for the proposed prioritized experience replay in this paper is to prioritize the trajectories where the model has the biggest misunderstanding of the corresponding scenarios, and therefore to prioritize long-tail scenarios. The final training pipeline is illustrated in Figure 6.

## 4 EXPERIMENTAL RESULTS

### 4.1 EXPERIMENTAL SETUP

**Dataset, simulator and metrics**. Training and experiments are conducted using the Waymo Open Dataset and the Waymax simulator. Waymax provides embedded support for reinforcement learning and diverse scenarios drawn from real driving data. It integrates with the Waymo Open Motion Dataset (WOMD), which offers 531,101 real-world driving scenarios for training and 44,096 scenarios for validation. Each scenario contains 90 frames of data. Specifically, WOMD v1.2 and the exact same metrics (off-road rate, collision rate, kinematic infeasibility, average displacement error (ADE)) from Waymax are used for benchmarking with the paper. Evaluations are conducted in both open-loop and closed-loop settings for a wider range of performance comparisons.

**Implementation Detail**. Models of various sizes are developed to quickly conduct ablation studies and assess final performance effectively. Raw observation takes ego vehicle, 15 nearest dynamic obstacles, 300 of closest roadgraph elements, traffic signals and position goal as input. The total size for each step observation is 8892 and feature extraction is applied to reduce the total size. SimDT(small) has 384 tokens for each element of $(s, a, g)$ pair, 10 blocks, 16 attention heads and in total 22.6 million parameters. SimDT(small) has 512 tokens for each element of $(s, a, g)$ pair, 15 blocks, 16 attention heads and in total 53.2 million parameters. Both models use context length with value 10, which means the causal transformer has access to past 10 $(s, a, g)$ pairs.

### 4.2 BENCHMARK COMPARISON

SimDT is evaluated both in open-loop and closed-loop settings for comprehensive comparison with various algorithms. During closed-loop evaluation, Intelligent Driving Model (IDM)(Treiber et al., 2000) is deployed as the simulated agent. SimDT achieves an Off-Road Rate of 3.36%, Collision Rate of 2.65%, Kinematic Infeasibility of 0.00%, and ADE of 6.73m. SimDT significantly outperforms them in collision rate and is second in off-road rate against other learning-based approaches. Compared with the same RL category method, SimDT demonstrates a substantial reduction in Off-Road Rate and Collision Rate than DQN by 45.2%. Similarly, the Collision Rate of SimDT shows 41% improvement over the Behavior Cloning (BC) model. Suggesting that our method is more effective at keeping the vehicle on the road and avoiding accidents. This improvement in safety-critical metrics highlights the robustness of SimDT in real-world driving scenarios.

| Agent | Action Space | Train Sim Agent | Off-Road Rate (%) | Collision Rate (%) | Kinematic Infeasibility (%) | ADE (m) | Route Progress Ratio (%) |
|---|---|---|---|---|---|---|---|
| Expert | Delta | - | 0.32 | 0.61 | 4.33 | 0.00 | 100.00 |
| Expert | Bicycle | - | 0.34 | 0.62 | 0.00 | 0.04 | 100.00 |
| Expert | Bicycle(D) | - | 0.41 | 0.67 | 0.00 | 0.09 | 100.00 |
| Wayformer (Nayakanti et al., 2022) | Delta | - | 7.89 | 10.68 | 5.40 | 2.38 | 123.58 |
| BC(Argall et al., 2009) | Delta | - | 4.14±2.04 | 5.83±1.09 | 0.18±0.16 | 6.28±1.93 | 79.58±24.98 |
| BC | Delta (D) | - | 4.42±0.19 | 5.97±0.10 | 66.25±0.22 | 2.98±0.06 | 98.82±3.46 |
| BC | Bicycle | - | 13.59±12.71 | 11.20±5.34 | 0.00±0.00 | 3.60±1.11 | 137.11±33.78 |
| BC | Bicycle(D) | - | **1.11±0.20** | 4.59±0.06 | 0.00±0.00 | **2.26±0.02** | 129.84±0.98 |
| DQN(Mnih et al., 2013) | Bicycle(D) | IDM | 3.74±0.90 | 6.50±0.31 | 0.00±0.00 | 9.83±0.48 | 177.91±5.67 |
| DQN | Bicycle(D) | Playback | 4.31±1.09 | 4.91±0.70 | 0.00±0.00 | 10.74±0.53 | 215.26±38.20 |
| SimDT(ours) | Bicycle | Playback | 3.52±0.26 | **2.69±0.10** | 0.00±0.00 | 7.14±0.63 | 106.47±2.8 |

Table 1: Closed-loop Benchmark. Performance evaluations are done against IDM simulation agents. Agents run without any termination conditions. Models report mean and standard deviation over 3 seeds. Action space is continuous unless denoted with D (discrete).

| Method | Train Sim Agent | Failure Rate (%) | Route Progress Ratio (%) |
|---|---|---|---|
| BC(Argall et al., 2009) | - | 4.35±0.27 | 99.00±0.39 |
| SAC(Haarnoja et al., 2018) | Playback | 6.66±0.44 | 77.82±8.21 |
| BC-SAC(Lu et al., 2023) | Playback | 3.35±0.31 | 95.26±8.64 |
| SimDT (ours) | Playback | 3.87±0.26 | 105.63±2.31 |

Table 2: Open-loop Benchmark. Failure rates (lower is better) of BC-SAC and baselines on different training/evaluation subsets. The failure rate is the percentage of the run segments that have at least one collision or off-road event. Models report mean and standard deviation over 3 seeds.

The open-loop evaluation (Figure 2) shows SimDT has comparable results with the state-of-the-art algorithm BC-SAC. BC again shows a solid performance with a low failure rate of 4.35% and a high route progress ratio of 99.00%. BC-SAC improves upon SAC, reduces the failure rate to 3.35% and increases route progress to 95.26%. Remarkably, our proposed SimDT method achieves a moderate failure rate of 3.87% while surpassing others with a route progress ratio of 105.63%, indicating not only effective navigation but also the potential for discovering more efficient routes. These results underline the importance of selecting appropriate training simulations and methods for optimizing autonomous navigation systems.

When compared to expert demonstrations, SimDT achieves competitive results in terms of safety metrics. Collision Rates are within the same magnitude as those reported by the experts. However, the ADE of SimDT is notably higher at 6.73m, which is approximately 6 meters away from the expert models. This suggests that SimDT learns a safe and feasible policy but is different from the expert recording. While the ADE for SimDT is higher than that of other imitation learning models, it is important to note that ADE alone may not capture the complete picture of driving performance. The emphasis on safety and kinematic feasibility by SimDT may contribute to a cautious driving style, which can result in a slightly higher ADE but with significantly safer outcomes.

## 4.3 ABLATION STUDY AND ANALYSIS

**Prioritized Experience Replay for Decision Transformer**. Compared to the baseline Decision Transformer model, integrating Prioritized Experience Replay results in a significant reduction of off-road incidents and collision rates by 26.1% and 5.5%, respectively (Table 3). Moreover, the enhanced Decision Transformer model achieves comparable performance using only 60% of the scenarios. The comparison of learning curves with and without Prioritized Experience Replay is presented in Figure 3. Multiple experiments are conducted with different initial seeds to ensure reproducibility. Three types of data are preferentially stored for PER (Figure 4). The initial cate-

| Agent | Off-Road Rate (%) | Collision Rate (%) | Kinematic Infeasibility | ADE (m) |
|---|---|---|---|---|
| DT(small) + 1 token prediction | 6.21±0.35 | 3.62±0.18 | 0.00±0.00 | 8.32±0.68 |
| DT(small) + PER + 1 token prediction | 4.59±0.31 | 3.42±0.15 | 0.00±0.00 | 7.95±0.69 |
| DT(small) + PER + OPA + 1 token prediction | 3.97±0.24 | 2.92±0.13 | 0.00±0.00 | 7.64±0.59 |
| DT(small) + PER + OPA + 3 token prediction | 3.82±0.25 | 2.65±0.11 | 0.00±0.00 | 7.52±0.55 |
| DT(small) + PER + OPA + 5 token prediction | 3.75±0.25 | 2.59±0.10 | 0.00±0.00 | 7.45±0.57 |
| DT(small) + PER + OPA + 7 token prediction | 4.14±0.30 | 3.33±0.11 | 0.00±0.00 | 8.19±0.59 |
| DT(median) + PER + OPA + 3 token prediction | 3.52±0.26 | 2.69±0.10 | 0.00±0.00 | 7.14±0.63 |

Table 3: Ablation Study. PER is prioritized experience replay, OPA is online policy adaptation.

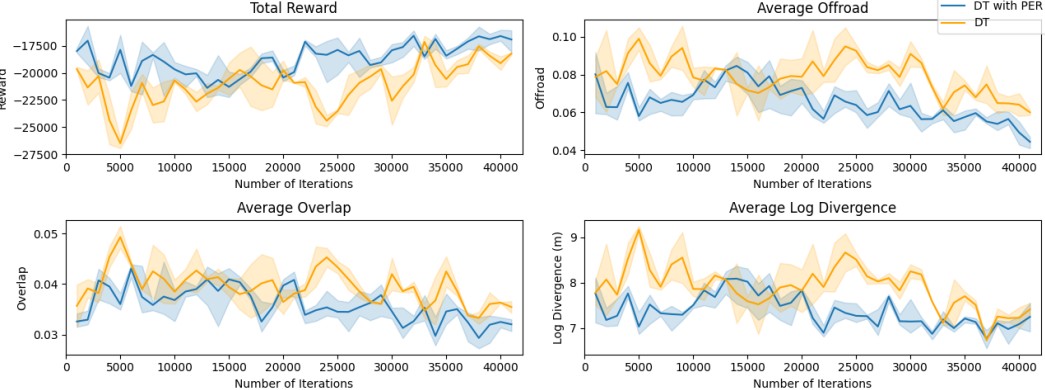

Figure 3: Comparison of learning curves for DT(small) settings with and without Prioritized Experience Replay. The metric for reward indicates that higher values are better, whereas for off-road incidents, overlap, and log divergence, lower values are better. Models report mean and standard deviation over 3 seeds.

gory encompasses instances wherein a discernible discrepancy arises between the predicted actions of the learning model and those executed by an expert. The second category pertains to scenarios wherein the cumulative action loss associated with a particular trajectory is substantially elevated, a phenomenon that predominantly transpires within the confines of rare encountered environmental conditions. The third category is representative of situations where trajectories indicative of sub-optimal online adaptation are documented, highlighting the model's challenges in identifying and rectifying suboptimal behaviors.

A key aspect of the Decision Transformer is its ability to model the relationship between state, action, and return-to-go ($g_t$). Given the same state observation, expert actions are associated with a higher $g_t$, and sub-optimal actions with a lower $g_t$. Ideally, DT should learn to act differently (e.g. choose an expert action, avoid driving out of the lane, or prevent collisions) based on the $g_t$ value. When a predicted action significantly diverges from the ground truth action given state observation and specific $g_t$, it indicates a misinterpretation of the environment. Intuitively, such misbehaved experiences need further reinforcement through the replay buffer.

From a high-level perspective, training on large-scale real-world data presents uneven scenario distributions. When the network begins to generalize common scenarios, rare scenarios like Figures 4(a) and 4(b) are undertrained. At this point, the network performs relatively well on common scenarios (low action loss) but not on rare scenarios (high action loss). Consequently, the replay buffer starts storing more of these rare scenarios. Criterion 2 is applied with accumulated action loss to prioritize such scenarios. Figure 4(c) illustrates a policy rollout during online adaptation where sub-optimal actions cause changes in $g_t$. What is important is the record of the corresponding $g_t$ with respect to action, where expert actions and sub-optimal actions are respectively given positive and negative rewards. eg. Figure 4(c) shows the ego vehicle driving out of the lane with a recorded decrease in $g_t$ value -2. It is crucial for the neural network to understand which actions cause a -2 value drop in $g_t$. These suboptimal state-action-return sequences are also important as expert driving data doesn't contain such behavior. The next time the agent encounters a similar scenario, the correct $g_t$ will help prevent out-of-lane actions.

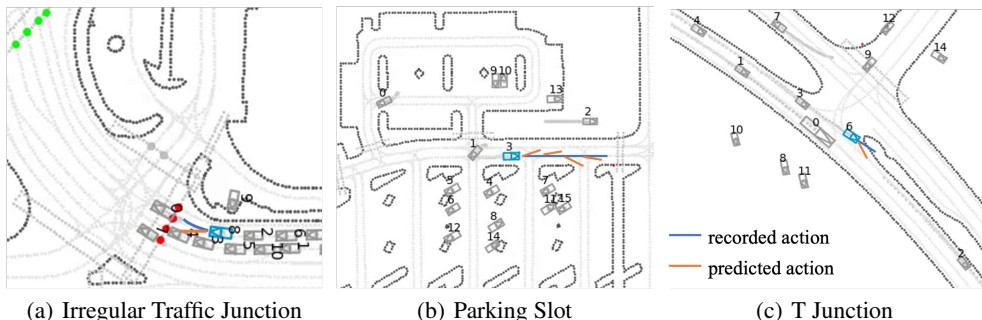

(a) Irregular Traffic Junction       (b) Parking Slot       (c) T Junction

Figure 4: Illustration of Data Selection for Prioritized Experience Replay: 4(a) is chosen because of its uncommon expert behavior that needs to slow down while steering to the right to keep lane. 4(b) illustrates a rare parking situation and was picked because it had the most mistakes when looking at the whole series of actions. 4(c) is kept as the suboptimal (collision) action taken in that situation was not reproduced given the corresponding low return-to-go.

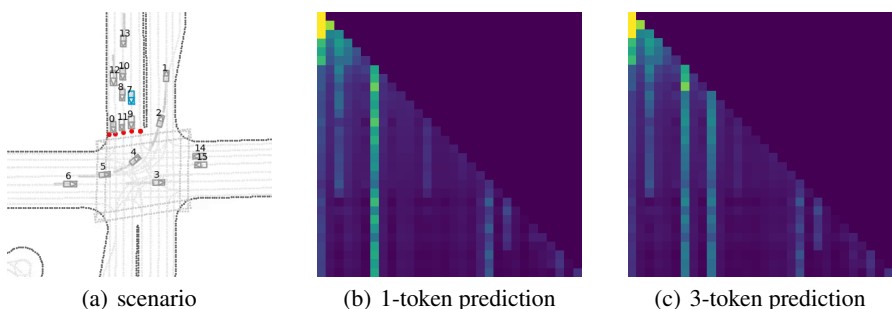

(a) scenario       (b) 1-token prediction       (c) 3-token prediction

Figure 5: Attention map comparison for single-token and multi-token prediction. Multi-token prediction network has a more diverse attention field. It indicates that multi-token prediction is less susceptible to immediate contextual patterns.

**Multi-token Decision Transformer**. In ablation study detailed in Table 3, we compare the performance of single-token prediction against multi-token predictions. The 3-token and 5-token SimDT variants demonstrate improvements of 3.78% and 6.80% in off-road, 8.56% and 11.3% in collision rates respectively. This improvement underscores the importance of considering future actions, influenced by real-world vehicle dynamics like inertia and momentum, which single-token models may overlook due to their focus on immediate contexts. The attention map in Figure 5 shows that multi-token prediction has wider attention fields towards long-term context. However, extending predictions to 7 tokens resulted in slight performance degradation, suggesting existing difficulties in trajectory stitching for larger sequences modeling by Decision Transformer. These results indicate that while multi-token prediction models offer a more nuanced understanding of environmental interactions, the optimization of larger token prediction poses a challenge that needs further study.

## 5 CONCLUSION

We present SmiDT, a novel sequence modeling-based reinforcement learning approach for closed-loop driving and complex real-world driving scenarios. Our fully online imitative Decision Transformer pipeline is adept at handling diverse data distributions found within extensive driving datasets, ensuring wide applicability and robustness. By implementing a multi-token Decision Transformer that integrates receding horizon control, we improve the model's ability to predict over longer horizons and extend its attention span across broader contexts. Furthermore, the incorporation of prioritized experience replay within our framework enhances the sample efficiency of training, allowing for more effective learning from large-scale datasets. Our work can also benefit other real-world robotics tasks that demand sample-efficient learning from expert demonstration and online adaptation.

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

# APPENDIX

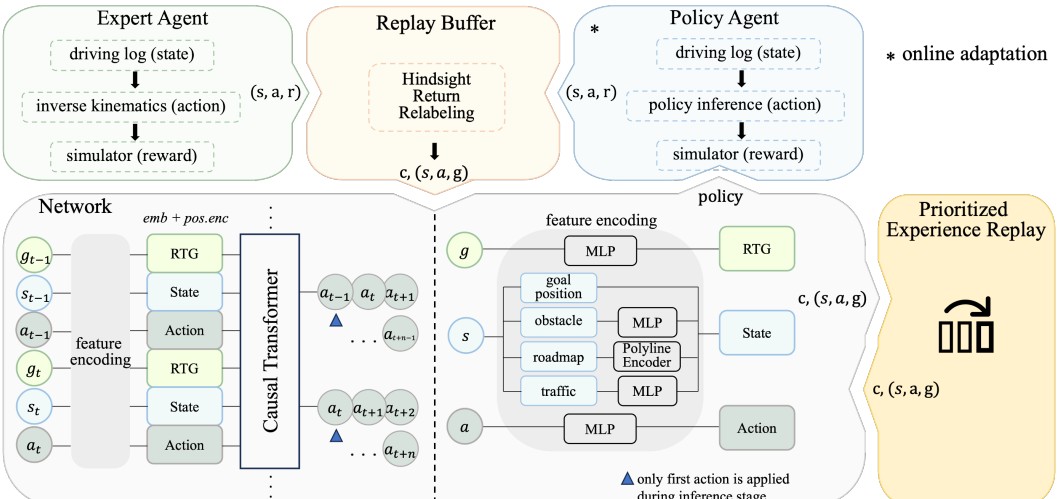

Figure 6: The general training procedure of SimDT. Real-world driving scenarios are reproduced in the simulator and the produced transitions $(s, a, r)$ are recorded to replay buffer. HindsightReturnRelabeling is performed to transform transitions to state-action-return pairs $(s, a, g)$. During training, prioritized experience replay is performed for sample efficiency. Finally, online adaptation is used for better performance.

## A. METRICS

The Off-road Rate indicates the percentage of whether the vehicle is driving within the road boundaries, with any deviation to the right of the road's edge considered off-road.

The Kinematic Infeasibility Metric is a binary metric that assesses whether a vehicle's transition between two consecutive states is within predefined acceleration and steering curvature limits, based on inverse kinematics.

Average Displacement Error (ADE) calculates the mean euclidean distance between the vehicle's simulated position and its logged position at corresponding time steps across the entire trajectory.

Route Progress Ratio calculates the proportion of the planned route completed by the vehicle, based on the closest point along the path at a given time step. The Route Progress Ratio feature is not released yet and the benchmark in this paper will skip this metric.

## B. CHOICE FOR IMITATION REWARD LOG_DIVERGENCE THRESHOLD

The log_divergence is the distance between the logged position and the actual agent position. The key here is to reward the system when the agent is close to expert human actions. When performing real-world human demonstration through inverse kinematics, the log_divergence is close to zero. The reason to set it to 0.2 is to reward policy agents. However, we consider 0.2 to be the maximum that we can set as values bigger than 0.2 can lead to a collision and out-of-lane actions which should not be rewarded. In general, we do not see a significant difference when varying values from 0.05 to 0.2. The reason can be 1. Change of threshold values has no impact on reward functions during the offline reinforcement learning phase. 2. Values under 0.2 are still relatively small for agents to follow during the online exploration phase.

## C. COMPARISON WITH BC-SAC

Despite SimDT and BC-SAC both trying to combine imitation learning and reinforcement learning, they use different approaches to achieve this goal. The key for BC-SAC is to combine reinforcement learning cost with imitation learning cost during training. Our proposed imitative reinforcement

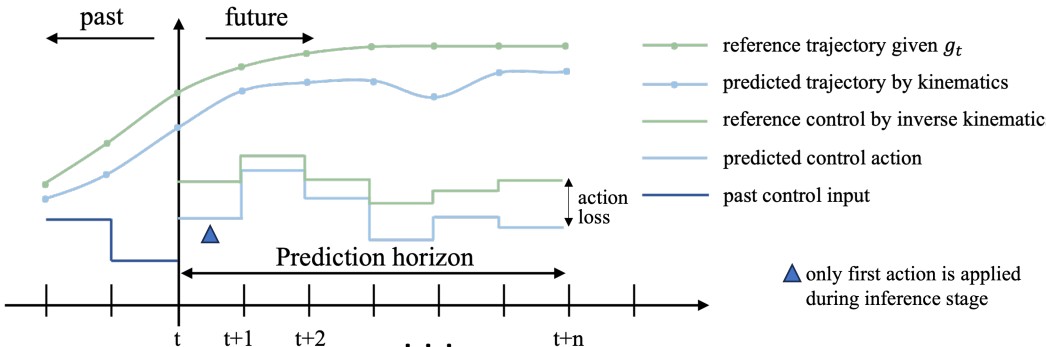

Figure 7: The general training procedure of SimDT. We minimize the action loss over a horizon to thereby minimize the deviation of the resultant predicted trajectory from the reference trajectory.

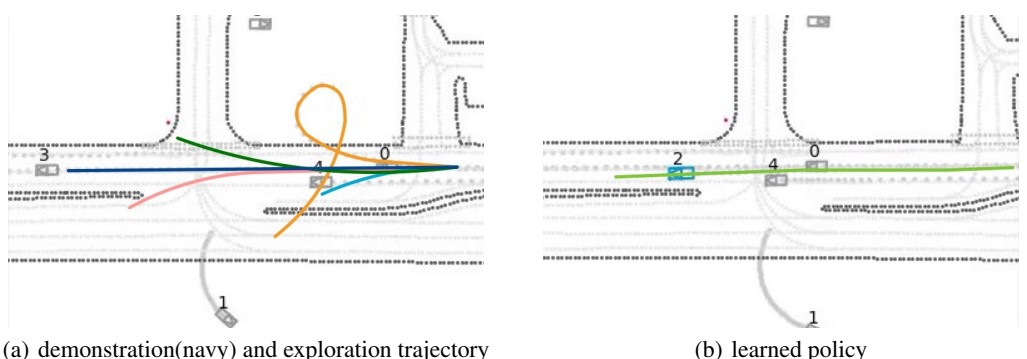

(a) demonstration(navy) and exploration trajectory      (b) learned policy

Figure 8: Illustration of how demonstration and exploration trajectory to learn a generalized policy.

learning pipeline is a reinforcement learning method that combines rewards for imitating expert demonstration.

D. COEFFICIENT FOR MULTI-TOKEN TRAINING

In the main body of the text, the modified loss function $L_{ma}$ is designed to enhance the network's learning capability regarding current step action predictions through the strategic use of coefficients $\alpha$, $\beta$, ..., and $\gamma$. The values that are adapted during training follow a decay setting as we think the near future token prediction is more important than further future tokens.

| Method | $\alpha$ | $\beta$ | $\gamma$ | $\delta$ | $\sigma$ | $\phi$ | $\omega$ |
|---|---|---|---|---|---|---|---|
| SimDT (1-token) | 1.00 | - | - | - | - | - | - |
| SimDT (3-token) | 0.80 | 0.40 | 0.20 | - | - | - | - |
| SimDT (5-token) | 0.80 | 0.30 | 0.15 | 0.10 | 0.08 | - | - |
| SimDT (7-token) | 0.80 | 0.25 | 0.12 | 0.10 | 0.08 | 0.06 | 0.05 |

