# OpenReview forum: "Sample-efficient Imitative Multi-token Decision Transformer for Real-world Driving"
_ICLR.cc/2025/Conference — ICLR 2025 Conference Withdrawn Submission_

### Official Review · Reviewer_o7X4 · 2024-10-27

**Soundness:** 2
**Presentation:** 2
**Contribution:** 2
**Rating:** 5
**Confidence:** 4

**Summary:**

This paper proposed SimDT, a DT-style method to combine imitation learning and online RL for driving. The main motivation is to handling the distribution shift problem in pure IL setting. The paper conduct experiments and ablation study to prove the efficiency of their method.

**Strengths:**

The paper is well written, the figures are nice. The ablation study is comprehensive.

**Weaknesses:**

The major concerns are the novelty and the performance of the proposed method. The authors proposed to combine online and offline RL training with decision transformer, which seems to be a quite straightforward combination of DT and online DT. Another drawback is that, the experiments results are not very strong and seems to be comparable with simple baselines. More recent baselines are missing.

**Questions:**

1. The claim ‘learning based agents face significant challenges when transferring knowledge from open-loop to closed-loop environment’ remains questionable to me. Since many recent advances in decision making follow the fashion of learning from offline dataset, and achieves superior performance in closed-loop control setting. In the experiment (table 1), the BC style method also achieves similar results with the proposed method.


2. The proposed multi-token prediction mechanism looks quite like action chunking proposed in recent works [1], which has been proved to be useful in many scenarios. Maybe some discussion and comparison are needed.


3. The baselines selection in the main experiments is not convincing (table 1&2). I think the baselines are too old (e.g., DQN, BC). Since the proposed method SimDT is based on DT style policy, I think it’s unfair to compare with some methods more than 10 years ago. Maybe other baselines like OnlineDT, or other recent works are needed as baselines.


4. I think the performance is not very strong. In the main experiments, BC + Bicycle(D) in table 1 and BC-SAC in table 2 seems to achieve comparable results with the proposed method.


5. Can you further explain on the metric “route progress ratio”? In Appendix A, it “calculates the proportion of the planned route completed by the vehicle”. Why it may achieve over 100%?



[1] Learning Fine-Grained Bimanual Manipulation with Low-Cost Hardware

---

### Official Review · Reviewer_YmUz · 2024-11-02

**Soundness:** 2
**Presentation:** 3
**Contribution:** 2
**Rating:** 5
**Confidence:** 4

**Summary:**

SimDT aims to address the distributional shift problem in closed-loop autonomous driving using a multi-token decision transformer. The paper proposes an online imitative learning pipeline and prioritized experience replay. The method is tested in both open-loop and closed-loop settings.

**Strengths:**

- This paper makes it easy for readers to grasp the main idea.
- The experiments are conducted in a closed-loop simulation.

**Weaknesses:**

- The author should compare more recent planning methods in autonomous driving [1]. SimDT outperforms the BC and DQN methods in the experiments section. However, I am curious whether a BC method with an appropriately designed network could potentially achieve better results. Additionally, suitable data augmentation [2] may more efficiently and simply alleviate the problem of OOD than RL-based methods. This is particularly relevant in the setting of autonomous driving, where there is an abundance of expert driving data.
- The fine-tuning pipeline requires model rollout in simulation. Due to the simulation domain gap, this may still lead to OOD problems in real-world applications.
- The author should discuss more related work in autonomous driving that utilizes multi-token prediction.

[1] Caesar, Holger, et al. "nuplan: A closed-loop ml-based planning benchmark for autonomous vehicles." *arXiv preprint arXiv:2106.11810* (2021).

[2] Bansal, Mayank, Alex Krizhevsky, and Abhijit Ogale. "Chauffeurnet: Learning to drive by imitating the best and synthesizing the worst." *arXiv preprint arXiv:1812.03079* (2018).

**Questions:**

- Are the model's output actions smooth during the closed-loop simulation? Why did you choose to supervise the actions using inverse dynamics, which differs from the commonly used waypoint or trajectory-level planning?
- Does the design of the reward influence performance ?

---

### Official Review · Reviewer_RH3z · 2024-11-03

**Soundness:** 1
**Presentation:** 1
**Contribution:** 1
**Rating:** 3
**Confidence:** 3

**Summary:**

This paper presents SimDT, a reinforcement learning framework for sequence modeling in interactive driving scenarios. The authors finetune a policy learned from the offline Waymo Open Motion Data using reinforcement learning in the Waymax simulator, using penalties for collision and going off the road. They evaluate their model on the Waymo Open Sim Agent Challenge (WOSAC).

**Strengths:**

Significance: the idea of using RL to improve transfer to closed-loop settings is innovative for improving sim agents.

**Weaknesses:**

While the concept of using reinforcement learning (RL) to improve transfer in closed-loop settings is innovative in the field of driving, the results presented in this paper are unconvincing. Additionally, the paper includes several unsupported and potentially incorrect claims. The following issues need to be addressed to improve the validity and contributions of this work.

**Major comments** (in order of importance)
1. Unsupported claims on performance gains. In the closed-loop evaluation, the authors claim that SimDT improves upon DQN by 45.2% in Off-Road Rate and Collision Rate and achieves a 41% improvement over a Behavior Cloning (BC) model. However, these performance improvements cannot be found in Table 1, and the actual improvements observed in Table 1 are much more modest (e.g., ~0.2% for Off-Road Rate compared to DQN, and about 2% for Collision Rate over BC). Misreporting these performance gains in the abstract and main text overstates SimDT’s effectiveness.
2. Lack of comparison with competitive baselines. A meaningful benchmark for SimDT would include a comparison to the Waymo Open Sim Agent Challenge (WOSAC) leaderboard (https://waymo.com/open/challenges/2024/sim-agents/), which includes the state-of-the-art for closed-loop agent realism and performance on the Waymo Open Motion Dataset. Evaluating SimDT against these established models would provide a clearer understanding of its strengths and limitations relative to current state-of-the-art baselines (as opposed to BC-SAC, which is not SOTA).
3. Missing information on dataset and evaluation. Tables 1 and 2 lack critical details: there is no information about the number of scenes trained and evaluated on, what percentage of the scenarios is used in practice? This makes it hard to interpret the results.
4. Misinterpretation route progress metric. The authors suggest that SimDT’s route progress ratio of 105.63% demonstrates the discovery of more efficient routes. However, a ratio above 100% does not necessarily mean a more efficient route; rather, it may simply indicate that the vehicle overshot the destination (e.g. by driving faster than the logged trajectory) or took a longer path. This metric interpretation, as outlined in the Waymax paper (https://arxiv.org/abs/2310.08710; page 5, Section 3.4), does not support the authors' conclusion and could be misleading to readers.
5. Slightly misleading comparison to expert performance. The authors claim that SimDT’s safety metrics are comparable to those of expert demonstrations, with Collision Rates "within the same magnitude" as expert results. However, the expert Off-Road and Collision Rates are significantly lower at 0.41% and 0.67%, respectively, compared to SimDT’s 3.52% and 2.69%. These differences should be put into context, as small percentage differences can have large practical impacts on safety in driving.
6. Claims on safety and ADE without evidence. The claim that SimDT's focus on safety and kinematic feasibility leads to a cautious driving style with a slightly higher average displacement error (lines 367-368) lacks empirical support.
7. Claims of sample efficiency without supporting information. Although the method is described as "sample-efficient," no information is provided about the training dataset size, RL training time, or computational resources. These details are important for substantiating claims of efficiency and should be included.

**Minor comments** (that did not impact my score)
- Line 477: "SmiDT" should be corrected to "SimDT."
- “Open-loop” is commonly used to describe settings where no feedback is provided, not specifically related to the behavior of other agents. I would suggest to clarify this to avoid misunderstanding.

**Questions:**

See my questions in "Major comments" above

---

### Official Review · Reviewer_hKsF · 2024-11-04

**Soundness:** 2
**Presentation:** 2
**Contribution:** 1
**Rating:** 3
**Confidence:** 3

**Summary:**

The paper addresses a set of important problem in self-driving: generalization to test-data distribution. Authors suggest that current methods are trained in open-loop scenarios and fail to generalize to closed-loop scenarios. In order to address this problem authors proposed 3 improvements:
1. A multi-token decision transformer.
2. An online reinforcement learning approach which transitions from offline traininging to online training to allow exploration of new scenarios.
3. A new scheme for sampling from replay buffer to prioritize scenarios where their policy is not performing well.

The authors validated and demonstrate the effectiveness of their approach through experiments on real-world datasets and ablation studies.

**Strengths:**

1. The paper addresses an important problem in autonomous driving.
2. All the experiments are conducted on the real-world Waymo dataset.
3. The authors show ablation studies to motivate their proposed improvements of the multi-token decision transformer, imitative RL pipeline, and prioritized experience replay.

**Weaknesses:**

1. The contributions seem weak, and the baselines are significantly outdated with the latest methods.
  - Authors compare against methods like DQN (Minh. 2013) and BC (Argall, 2009). These are very old methods.
  - There have been many new RL algorithms like Rainbow DQN, TD3BC, CQL, and AWAC.
  - Many transformer-based approaches like Point Transformer V3 Extreme, MotionTransformer, etc.

*Suggestion*: Please add the latest baselines. Baselines from the 2024 Waymo Open Dataset Challenge are a good start.

2. Although the paper is well-written, it lacks technical rigor and is hard to follow.
  - What exact problem are authors solving? From my understanding, the problem is vaguely introduced only in the introduction.
  - The paper does not clearly explain where and how current methods fail.
 -  Fig 1 is unclear. The purpose of outside black lines is not clear. I assume the blue lines are the new trajectory sampled by the authors' method.

*Suggestion*: Please add a problem formulation section. Give some examples of how single-token Decision Transformers fail.

3. The method seems credible, but it is heuristically put together.
  - "The overall online imitative reinforcement pipeline is essential to achieve the greater data-distributed policy" How does author's method  lead to greater data-distributed policy.
  - R_{imitation} is not clearly explained.
  - Where is R_{imitation} used ? I do not see it in Algorithm 1.
  - Switching from offline to online learning seems to have been heuristically chosen. The motivation behind the 0.5 ∗ num scenarios is unclear.

*Suggestion*: I suggest that the authors rewrite the method section to add technical rigor. Each design decision needs to be clearly explained.

4. The math is not clearly and rigorously defined:
  - What are $a$, $s$, $g$, $\pi$ variables? Whether they are scalars, vectors, matrices, or function mapping is unclear.
  - It is not clear where loss functions L_a and L_ma are used.

My recommended score for the paper is based on the lack of up-to-date baselines and technical rigor. In my opinion, the paper needs a significant amount of work to be accepted.

**Questions:**

1.  What are $a$, $s$, $g$, $\pi$ variables? It is not clear if they are scalars, vectors, matrices, or function mapping.
2.  Where are loss functions L_a and L_ma used?
3.  Where is R_{imitation} used ? I do not see it in Algorithm 1.
4. How did authors arrive at rewards for off-road = -2 and rewards of overlap = -10?
5. How do authors decide to switch from offline learning to online learning in Algorithm 1?
6. How are $\alpha$ and $\beta$ picked in Eq 2?

---

### Official Review · Reviewer_PZd3 · 2024-11-04

**Soundness:** 1
**Presentation:** 3
**Contribution:** 1
**Rating:** 3
**Confidence:** 4

**Summary:**

The paper proposes SimDT, a decision transformer architecture for autonomous driving. The proposed method leverages prioritized experience replay for efficient learning. It also combats distribution shift problem in the RL problem setup. The result shows big improvement over SOTA on collision rate.

**Strengths:**

1. The introduction of multi-token prediction in a decision transformer framework is interesting and may help with the realtimeness of the algorithm.
2. The writing is easy to follow, and the author’s method is clearly explained.
3. Experiments include representative SOTA methods and ablation studies demonstrating the necessity of individual components in open and closed-loop settings.

**Weaknesses:**

1. Overall lack of novelty. There is little novel components introduced in the paper. Multi-token prediction has been explored in NLP and RL; PER is classical and nearly 10 years old; the novelty in combating distribution shift in imitation learning is also unclear.
2. Flaw in experiment design: since the authors’ main argument is that their proposed method has the lowest collision rate, is it possible that this simply comes from the fact that they assigned collision with a very high penalty in the RL? According to equation 5, R_{overlap} = -10 in the method. I don't see any related experiment or discussion to remove this doubt.
3. No significant improvement overall compared to SOTA: given the unaddressed flaw mentioned above, plus the fact that the SimDT cannot consistently outperform SOTA on most if not all of the metrics, I think it is valid to suspect that even the low collision rate performance of SimDT might not have come from the robustness of the algorithm itself, but simply from reward engineering.
4. Lack of experiment for “sample-efficient”: I think this part of the title requires a controlled study (fixed amount of data or training FLOPs) to provide empirical results to justify.

**Questions:**

1. More justifications for their claimed performance on collision rate and other metrics invariant of reward engineering (see above)
2. More controlled study on sample efficiency (see above)
3. Do the authors have more information to add on the overall novelty of the approach?
4. Could there be other set of metrics, preferably commonly used, that can further help evaluate all the listed methods?

Technicality:
1. What specific information does Figure 3 intend to show? I find related discussions insufficient.
2. Table 3 can be cleaner with better caption and bolding.

---

### Official Review · Reviewer_EwHs · 2024-11-08

**Soundness:** 3
**Presentation:** 3
**Contribution:** 2
**Rating:** 1
**Confidence:** 4

**Summary:**

To address the data distribution shift problem when applying supervised-learning or offline RL based behavior model to the closed-loop simulation environment, this paper proposes SimDT, an online imtative learning transformer. The decision transformer is multi-token and equipeed with prioritized experience replay. During testing, receding horizon control is used. Hindsight relabelling is used to assign reward to the data.

**Strengths:**

1. The paper uses decision transformer, with a set of practices in online RL (prioritized replay buffer), to address the closed-loop planning task in Waymax.

**Weaknesses:**

1. The multi-token transformer is not novel at all. In the task of simulation agent, multi-token transformer is a standard practice [1,2,3,4] (should note that the multi-token in sim agents is multiple tokens for agents at the same step, instead of multiple tokens for an agent). My overall idea is that multi-step prediction + recending horizong control is not surprising. In Waymo Sim Agent benchmark [5] and Waymax paper, using receiding horizon control on the "one-shot" model is a standard practice.
2. The combination of hindsight replay, prioritized replay buffer is promising. But they are not suprising and their benefits are expected.
3. Overall, my concern is that the paper lack of novelty. I personally don't prefer the paper putting a bunch of existing practices together and claims we improved the scores, without extensive study on why it works and what insights we can learn.


[1] MotionLM: Multi-Agent Motion Forecasting as Language Modeling

[2] KiGRAS: Kinematic-Driven Generative Model for Realistic Agent Simulation

[3] SMART: Scalable Multi-agent Real-time Motion Generation via Next-token Prediction

[4] Trajeglish: Traffic Modeling as Next-Token Prediction

[5] The Waymo Open Sim Agents Challenge

**Questions:**

1. Missing some relevant papers:

* "CtRL-Sim: Reactive and Controllable Driving Agents with Offline Reinforcement Learning" Using offline RL to learn multi-agent behavaior.
* The Sim Agent models I mentioned above.
* "Improving Agent Behaviors with RL Fine-tuning for Autonomous Driving" Using RL to finetune multi-agent behavior model.

---

### Note · Authors · 2024-11-13

I have read and agree with the venue's withdrawal policy on behalf of myself and my co-authors.